# Visual Disfunction due to the Selective Effect of Glutamate Agonists on Retinal Cells

**DOI:** 10.3390/ijms22126245

**Published:** 2021-06-10

**Authors:** Santiago Milla-Navarro, Ariadna Diaz-Tahoces, Isabel Ortuño-Lizarán, Eduardo Fernández, Nicolás Cuenca, Francisco Germain, Pedro de la Villa

**Affiliations:** 1Department of System Biology, University of Alcalá, 28871 Madrid, Spain; santiago.milla@edu.uah.es; 2Instituto Ramón y Cajal de Investigación Sanitaria (IRYCIS), 28034 Madrid, Spain; 3Bioenginering Institute, Miguel Hernández University and CIBER-BBN, Elche, 03002 Alicante, Spain; adiaz@umh.es (A.D.-T.); e.fernandez@umh.es (E.F.); 4Department of Physiology, Genetics and Microbiology, University of Alicante, 03690 Alicante, Spain; isortliz@ua.es (I.O.-L.); cuenca@ua.es (N.C.)

**Keywords:** NMDA, kinate, excitoxicity, bipolar cell, amacrine cell, retinal ganglion cell, multielectrode recording, pERG, optomotor test

## Abstract

One of the causes of nervous system degeneration is an excess of glutamate released upon several diseases. Glutamate analogs, like N-methyl-DL-aspartate (NMDA) and kainic acid (KA), have been shown to induce experimental retinal neurotoxicity. Previous results have shown that NMDA/KA neurotoxicity induces significant changes in the full field electroretinogram response, a thinning on the inner retinal layers, and retinal ganglion cell death. However, not all types of retinal neurons experience the same degree of injury in response to the excitotoxic stimulus. The goal of the present work is to address the effect of intraocular injection of different doses of NMDA/KA on the structure and function of several types of retinal cells and their functionality. To globally analyze the effect of glutamate receptor activation in the retina after the intraocular injection of excitotoxic agents, a combination of histological, electrophysiological, and functional tools has been employed to assess the changes in the retinal structure and function. Retinal excitotoxicity caused by the intraocular injection of a mixture of NMDA/KA causes a harmful effect characterized by a great loss of bipolar, amacrine, and retinal ganglion cells, as well as the degeneration of the inner retina. This process leads to a loss of retinal cell functionality characterized by an impairment of light sensitivity and visual acuity, with a strong effect on the retinal OFF pathway. The structural and functional injury suffered by the retina suggests the importance of the glutamate receptors expressed by different types of retinal cells. The effect of glutamate agonists on the OFF pathway represents one of the main findings of the study, as the evaluation of the retinal lesions caused by excitotoxicity could be specifically explored using tests that evaluate the OFF pathway.

## 1. Introduction

An excess of glutamatergic stimulation in the nervous system is at the origin of many neurodegenerative diseases in mammals [1,2]. The toxicity generated by excessive glutamate is developed through the activation of ion channels. Different studies show that an increase in intracellular calcium concentration is associated with the hyperactivity of excitatory amino acids [3], but not with that of non-excitatory [4], revealing the important role that calcium ions play in excitotoxicity. N-methyl-DL-aspartate (NMDA) has been described as the glutamate analog that shows the greatest potency in increasing calcium influx and inducing neurotoxicity, and NMDA-R has been described as the receptor responsible for mediating excitotoxicity [4]. In accordance with this, the use of glutamate antagonists has been shown to provide neuroprotection in animal models of neuronal injury [5]. Kainic acid (KA) is the other major glutamate agonist with an important role in neurotoxicity. As with NMDA, neuroprotection has been demonstrated through the use of its antagonists [6]. In the case of KA, a low concentration of this molecule is capable of causing an increase of intracellular calcium and the death of neural cells, without significant depolarization [7].

Physiologically, glutamate acts through NMDA or KA receptors, so in order to properly assess the excitotoxic effect of glutamate, it must be induced by both pathways. Furthermore, as the activation of NMDA receptors by glutamate requires the cell to be depolarized, the joint action of KA could depolarize it, inducing a greater effect of glutamate on NMDA receptors. As proof of this, a significant protection against neuronal death has been demonstrated by the direct antagonism of NMDA and AMPA receptors [5]. Their neuroprotective mechanism could work by antagonizing the cellular calcium influx [8] or through a chelating effect on intracellular calcium [9].

The mammalian retina has been proven to be a useful model for the study of neuronal excitotoxicity. In mice, glutamate excitotoxicity mainly affects the inner retinal layers [10,11,12], location of bipolar cells, conventional and displaced amacrine cells, and ganglion cells [13,14], which are sensitive to glutamate agonists acting on ionotropic glutamate receptors [11]. However, not all retinal cell types are susceptible to excitotoxicity to the same extent; the excitotoxicity induced by NMDA seems to have a strong effect on amacrine cells, a mild effect on bipolar cells, and no effect on photoreceptors [15]. In fact, not even different types of retinal ganglion cells (RGC) have shown the same sensitivity to excitotoxicity. Those RGC with a large soma seem to be more resistant to NMDA excitotoxicity than small RGC [16].

In a previous study of our group, a co-dose of 30 mM NMDA and 10 mM KA was shown to induce a deleterious effect on the inner retina [12]. Electroretinogram (ERG) results showed a significant decrease of the retinal “b” wave amplitude, both in scotopic and photopic conditions. However, the “a” wave amplitude did not change significantly, indicating the preservation of photoreceptors. Histologically, although no effects in the outer nuclear layer were observed, a significant thinning on the inner retinal layers was reported, indicating that NMDA and KA were able to induce a harmful effect on bipolar, amacrine, and ganglion cells. In addition, anterograde tracing of the visual pathway after NMDA and KA injection showed the absence of RGC projections to the contralateral superior colliculus and lateral geniculate nucleus [12]. However, the way in which the cell death process occurred seemed to depend on the magnitude of the excitatory response generated by the inoculated dose [17]. Thus, the dose is so important that it not only determines the type of death, but whether it can induce survival or death [18,19,20,21].

As different cell types exhibit a different response to glutamate toxicity because of their differential composition of glutamate receptors, in the present study, we aimed to determine the cells and pathways that best resist glutamatergic excitotoxicity. We performed new experiments using new lower doses of glutamate agonists, below the level that produces a deleterious effect. Our approach included in vivo studies to assess the functionality of the inner retina, using recording pattern ERG [22], and the spatial visual acuity, using the optomotor test [23]. We also performed ex vivo electrophysiological recordings using a multi-electrode array and immunohistological analysis of different retinal cell types.

## 2. Results

### 2.1. Immunostaining

The histological effects that glutamatergic hyperstimulation caused in the retina were studied by immunohistochemistry using specific antibodies against Syt2b (ZNP-1/synaptotagmin 2), calbindin, PKCα, Brn3a, TH, Dab1, ChAT, and calretinin.

In the control retinas injected with PBS, Syt2b antibodies showed complete OFF cone bipolar cells, with stronger immunostaining of the bipolar terminals in the OFF sublamina of the inner plexiform layer (IPL), corresponding to axon terminals of type 2 cone bipolar cells. A fainter immunostaining of some bipolar terminals in the ON sublamina could be also observed, corresponding to type 6 cone bipolar cells stratification (Figure 1A,D,G, arrowheads). Immunolabeling with calbindin antibodies showed dendrites and axon terminals of horizontal cells in the OPL (Figure 1D,G). In response to 1:0.3 mM NMDA/KA injection, there was a disorganization of the IPL, where most of the Syt2b OFF terminals were lost, and only some bipolar terminals could still be identified in the IPL ON stratum (Figure 1B,E,H, arrowheads). Just a few distorted axons remained in the stratum OFF (Figure 1H), but the axon terminals of the ON cone bipolar cells could still be identified in the IPL ON stratum (Figure 1H arrows). Although the immunoreactivity of the OFF bipolar cell bodies and terminals decreased at this dose, immunoreactivity to horizontal presynaptic synaptosomes was visible at the OPL with a normal morphology, and just some sprouting could be observed. In addition, some dendritic terminals at the OPL were conserved, and it is reasonable to think that at least a portion of the cone bipolar cells remained alive. Horizontal cells (red) were maintained at this dose (Figure 1E,H), but small sprouts of horizontal and cone bipolar cells (green) towards the outer nuclear layer (ONL) began to appear. In response to 10:3 mM NMDA/KA injection, a strong thinning of the retinal inner layers was observed (Figure 1C,F). The INL, IPL, and GCL layers showed a drastic thickness decrease, while for the IS, ONL, and OPL layers, the thicknesses were maintained, indicating that the photoreceptors were not affected by the NMDA and KA injection. Cone bipolar cells immunostained with the Syt2b antibody were almost completely lost, and only a few ON terminals remained in the IPL (Figure 1C,F, arrowheads). Despite the major bipolar cell loss and inner layers’ disorganization, the horizontal cells remained (Figure 1F,I), but they projected long processes towards the ONL (Figure 1I,J, arrows), probably in search of new connections due to the loss of the inner layers. 

In the case of ON rod bipolar cells, PKCα labeling showed that they were slightly shortened in response to 1:0.3 mM NMDA/KA injection, compared with the control retina (Figure 2A,B,H,I). With higher doses of NMDA/KA (Figure 2C,J–L), the shortening of rod bipolar cell axonal processes was evident, accompanied by a global inner retina thinning. Nevertheless, in contrast with what happened to the OFF bipolar cells, here, the ON rod bipolar cells still maintained some of their terminals at the IPL. This fact suggests a bigger effect of the NMDA/KA mixture in the OFF pathway. When using the 10:3 mM NMDA/KA dose, sprouts of the rod bipolar cell dendrites towards the outer retina were also clearly observed (Figure 2C,K, arrow). RGC (Figure 2A–F, red) also seemed to be affected by increasing concentrations of NMDA/KA, and their gradual loss accompanied the thinning of the inner retina previously described.

Regarding the state of amacrine cells after treatment, in response to 1:0.3 mM NMDA/KA injection, the labeling of amacrine cells showed a decrease in the labeling intensity of AII and dopaminergic amacrine cells, and an evident loss of calretinin and ChAT amacrine cells (Figure 3). At the IPL, dopaminergic amacrine cells formed a plexus that synapsed with the bodies of the AII amacrine cells, which seemed to be maintained at this dose of excitotoxic agents (Figure 3J,K). In contrast, the plexus of starburst amacrine cells disappeared almost completely in both the OFF and ON sublayers of the IPL, and the pattern of the three levels of stratification observed with calretinin labeling was not visible anymore (Figure 3D,E,G,H,M,O). In response to 10:3 mM NMDA/KA injection, the amacrine loss was dramatic and there was almost no AII, dopaminergic, starburst, or calretinin-labeled amacrine cell remaining. As a consequence, their plexus at the IPL also disappeared almost completely (Figure 3C,F,I,L,P).

### 2.2. Retinal Multielectrode Recording

The response properties of different types of RGC (ON, OFF and ON/OFF) was analyzed both in the control animals and in those injected with 10:3 mM NMDA/KA. In addition to the decrease in the total number of recorded RGC in the NMDA/KA injected retinas, a change in the relative proportions is observed. From a total of 237 RGC recorded from four control eyes, 37.5% (*n* = 89) were ON, 14.7% (*n* = 35) were OFF, and 47.6% (*n* = 113) were ON/OFF while in the NMDA/KA injected eyes, from a total of 50 RGC recorded from four animals, 96% (*n* = 48) were ON, 4% (*n* = 2) were OFF, and no ON/OFF responses were recorded. These data indicate a statistically significant increase in the proportion of ON-type RGC and the absence of ON/OFF type RGC (Figure 4A). The possibility that ON/OFF type RGC loses the OFF response should not be discarded (see discussion). The transient or sustained responses of RGC were also analyzed. All types of responses were observed in the control group, while in the NMDA/KA injected group, we were not able to observe any sustained OFF nor ON/OFF response (Figure 4B). In summary, there was a global impairment of the retina with special incidence in the OFF responses. 

Light sensitivity was also analyzed in the ON-type RGC. Sensitivity to the light stimuli of increasing intensities differed between the control and NMDA/KA injected eyes. RGC from the control retinas were more sensitive to light stimuli of any tested intensity (Figure 5A). A proportion of 29% of RGC in the control retinas were sensitive to 6.2 cd·s/m^2^, while in the NMDA/KA group, just 4% of cells were sensitive to such a light intensity. For any light intensity of a higher magnitude, statistically significant differences were observed between the control and NMDA/KA groups (chi-square *p* < 0.001). Furthermore, a significant reduction in firing frequency was observed in the NMDA/KA group (Mann–Whitney U), both during the light stimuli and basal activity (Figure 5B). Response latency was also analyzed in RGC from both experimental groups (Figure 5C) and a statistically significant increase was observed in the NMDA/KA injected group (Mann–Whitney U). Finally, a statistically significant decrease of the RGC receptive field was observed in the NMDA/KA injected group when compared with the control group (Mann–Whitney U; Figure 5D).

We further tested the directional selectivity of the recorded RGC. Cells with/without directional selectivity could be recorded in both experimental groups (Figure 6). The proportion of RGC with directional selectivity did not differ significantly from the control group to the NMDA/KA injected group. Just 18 out of 208 RGC (≈9%) of the control retinas showed a direction index >0.5, while 9 out of 46 RGC (≈20%) in the NMDA/KA injected group showed a direction index >0.5. However, an apparent different response accuracy was observed in the directional sensitivity between both experimental groups (Figure 6), with the response being quite accurate in the control group and quite broad in the NMDA/KA injected group, although the degree of accuracy was not statistically analyzed because of the huge variability of the cell responses. 

The single cell results obtained by the multielectrode recordings suggested that the injury caused by the injection of NMDA/KA affected the functionality of the RGC that remained alive, with a higher effect on the OFF response.

### 2.3. Pattern Electroretinography

A pattern electroretinogram (pERG) allows for studying the functionality of the ganglion cell population by analyzing the response of the whole retina to changes of contrast through checkerboard stimulation (Figure 7A). The effect of the different NMDA/KA doses on the wave amplitudes was studied at four spatial frequencies (0.08, 0.12, 0.17, and 0.31 cpd). pERG recordings were obtained from the same group of animals before and one week after the injection of the excitotoxic mixture (1:0.3, *n* = 7; 3:1, *n* = 7; 10:3, and *n* = 4, mM NMDA/KA) into the right eye. Three characteristic waves (N35, P50, and N95) were recognized in control/preinjection recordings at the four different spatial frequencies tested. However, one week after the injection of 3:1 mM NMDA/KA, the component N35 was not clearly identified, and P50 and N95 showed an increased latency and decreased amplitude compared with the preinjection recording, reflecting the functional damage of the ganglion cell population (Figure 7B). No statistically significant difference in the N35, P50, or N95 wave amplitudes were observed among the different spatial frequencies in the control experiment.

Significant differences between the left eye (PBS injected) and right eye (NMDA/KA injected) were observed when using excitotoxic concentrations above 3:1 mM NMDA/KA. The injection of 1:0.3 mM NMDA/KA did not induce any significant reduction in the N95 wave component amplitude for any spatial frequency (Figure 7C, left, two-way ANOVA, *p* = 0.2561), confirming that the functionality of the ganglion cell population was not significantly affected by this dose. However, in the 3:1 mM NMDA/KA injected eye, the decrease in the N95 wave component was statistically significant compared with the control eye for any spatial frequency (Figure 7C, middle, two way ANOVA, *p* < 0.0001). A reduction of ca. 70% for the N95 wave amplitude was observed. Bonferroni post-test analyses showed most significant differences for spatial frequencies of 0.08 and 0.12 cpd (*p* < 0.001 and *p* < 0.01, respectively). Likewise, the injection of 10:3 mM NMDA/KA induced a statistically significant decrease in N95 wave component amplitude between the injected and control eyes for any spatial frequency (Figure 7C, right, two-way ANOVA, *p* = 0.0197). 

### 2.4. Optomotor Test

To test the visual behavior after the injection of NMDA/KA, an optomotor test was carried out on the same animals in which the pERG was performed. Through this test, the mice’s eye and head movements were recorded when the animals followed, with their gaze, the moving vertical bars presented on the screens (Figure 8A). Different spatial frequencies (0.011, 0.022, 0.044, 0.088, 0.177, and 0.355 cpd) and contrasts (100, 50, 25, 10, and 5%) were explored. Gradual contrast (white to black or black to white) of the bars moving in both directions (clockwise and anticlockwise) allowed us to selectively test the function of the ON and OFF retinal pathways of the left (PBS injected) and the right (NMDA/KA injected) eyes (Figure 8B). The highest contrast sensitivity of the animals was observed for a spatial frequency of 0.088 cpd. Any higher or lower frequency showed a decrease in contrast sensitivity. Comparisons between the white to black gradients with the black to white gradients, perceived by the ON or OFF retinal pathways, did not show statistically significant differences in the control animals (two-way ANOVA, *p* = 0.4125). 

The optomotor test showed a clear correlation between the doses of injected excitotoxic agents and visual sensitivity. Injection of the lowest dose of NMDA/KA (1:0.3 mM) into the right eye did not affect the animals’ ability to detect moving bars in both clockwise and counterclockwise directions (Figure 8C, left). Specific stimulation of the ON and OFF pathways showed a similar sensitivity for each spatial frequency. Comparisons between the white to black gradients with the black to white gradients, perceived by the ON or OFF retinal pathways, did not show statistically significant differences in the 1:0.3 mM NMDA/KA injected retinas (two-way ANOVA, *p* = 0.2645). 

On the other hand, injection of the highest dose of NMDA/KA (10:3 mM) into the right eye prevented the animals from detecting counterclockwise bar displacement, while they were still able to identify clockwise bar movement, controlled by the PBS injected eye (Figure 8C, right). These results indicate that the excitotoxic agent injection induced visual deficiencies just in the preferred direction, as perceived by the retina of the damaged eye. 

The injection of 3:1 mM NMDA/KA into the right eyes did not affect the ability of the animals to detect moving bars in a clockwise direction, as the left retina was not damaged. However, it affected the detection of moving bars in a counterclockwise direction: they were not detected when black to white gradual bars were applied (stimulation of the OFF retinal pathway), while they were still detected when white to black gradual bars were used (stimulation of the ON retinal pathway) at 0.044, 0.088, and 0.177 cpd (Figure 8C, middle). Comparisons between the white to black gradients perceived by the ON retinal pathways for those of the spatial contrast showed statistically significant differences in in 3:1 mM NMDA/KA vs. PBS injected retinas (two-way ANOVA, *p* = 0.0192). 

All together, these results indicate a stronger damage effect of the excitotoxic agents on the OFF retinal pathway than on the ON retinal pathway, as supported by the electrophysiological recordings and the histological analysis. 

## 3. Discussion

The retinal excitotoxicity caused by the intraocular injection of NMDA and KA is capable of inducing the death of ganglion cells, amacrine cells, and bipolar cells, as these cells express said ionotropic glutamate receptors [24,25,26]. Likewise, the action of glutamatergic agonists produces massive disorganization of the inner retina, even affecting the outer retina. All of these effects cause an alteration of retinal functionality that translates into a decrease in visual acuity and difficulty in detecting stimuli in motion. Our results show that the histological changes that the retina undergoes as a result of glutamatergic overstimulation have a progressive loss of retinal functionality, which is reflected by the alteration of the electrophysiological properties of the ganglion cells and therefore of the information that contribute to visual centers.

Although previous works induced excitotoxicity by activation of NMDA or KA receptors in the retinal cells, given that both receptors are naturally activated by the physiological neurotransmitter glutamate, it seems logical to think that NMDA/KA coactivation must be the pathophysiological mechanism causing an excitotoxic effect on the retinal neurons during different nosological processes (ischemia, axonal compression, metabolic diseases, glaucoma, etc.). In a previous study [12], we demonstrated the deleterious effect of the joint intraocular application of both glutamatergic agonists. While the intraocular injection of NMDA alone needs a dose of 100 mM to produce a maximum lethal effect on ganglion cells, and a dose of 5 mM KA induces the death of ca. 50% of ganglion cells, and our present work shows that a concentration of 3 mM KA and 10 mM NMDA, when applied together, induces a much bigger effect on ganglion cell death than a single application. After joint treatment, only 20% of ganglion cells could be recorded in NMDA/KA injected eyes when compared with the control eyes. These data agree with some studies carried out in cultured retinal neurons, which are not affected by the stimulation of NMDA receptors alone, but are sensitive to the stimulation of non-NMDA receptors [27]. More specifically, it has been shown that KA is toxic to ganglion cells, but its effect is greater when injected together with NMDA. As the stimulation of the KA receptors achieves cellular depolarization and, therefore, sensitization of NMDA receptors—now free from the blockade by Mg—when activated together they induce a massive influx of Ca2 + and the consequent cell death.

In the present work, we used different doses of both glutamatergic agonists, injected intraocularly together, to try to elucidate the different sensitivities of different retinal cell types to increasing doses of NMDA and KA, and their impact on some visual functions. The excitotoxicity caused by the joint intraocular inoculation of NMDA and KA achieves the simultaneous activation of NMDA and non-NMDA receptors, which causes the destructuring of the inner plexiform layer and the loss of cells in the inner retina [12]. The present work shows how increasing doses of NMDA and KA induce the death of bipolar, ganglion, and amacrine cells, leading to huge structural changes of the inner retina. However, it also shows that the damaging effects are manifested in the outer retina, as evidenced by the fact that growing cell extensions appear to be emerging from horizontal cells, accompanied by the processes of some bipolar cells. Although it is not clear why this effect occurs, it could be that the injury suffered by ganglion and amacrine cells in the IPL causes the disconnection of bipolar cells and the retraction of their dendritic tree at this level, and induces the outgrowth of horizontal and bipolar cell processes in the external retina seeking some kind of reconnection. In this sense, the appearance of growth shoots in the dendritic tree of ganglion cells has been observed after an injury to its axon [28,29].

The described structural alterations of the inner retina lead to a decrease in cellular functionality that is more intense in the OFF than in the ON pathway of visual processing. Undoubtedly, this is due to a greater loss of cells that integrate the OFF pathway. In this sense, a progressive decrease in the labeling of amacrine AII cells, which carry information from the rod pathway to the cone pathway, is also notable as the NMDA/KA dose increases. 

The damaging effect of excitotoxic agents on retinal cells also results in a decrease in the firing frequency of ganglion cells, both in their basal activity and in their response to light stimuli, because of the increased latency between shots. A reduction in the size of the receptive field has also been observed. Different types of ganglion cells could show differences in sensitivity to excitotoxic agents [16] related to the expression level of the ionotropic glutamate receptor [30]. Large, alpha-like ganglion cells show a low expression of calcium-permeable glutamate receptors. However, ON/OFF direction-selective ganglion cells, and OFF ganglion cells show higher levels of expression for these receptors, which causes a greater effect of excitotoxic agents and a greater mortality of the cells. These data can explain our functional results, in which we see greater damage, but not are exclusive to the cells of the OFF pathway.

Our results show that after the injection of high doses of NMDA and KA, there is an absolute loss of the ON/OFF responses of the ganglion cells, as well as a significant decrease in the OFF responses. It cannot be ruled out that some of the ON cells that were registered after the effect of the excitotoxic agents are ON/OFF type cells that have lost the OFF response. Similarly, it is striking that even at these doses of excitotoxic agents, cells with a directional selectivity can still be seen. One explanation could be that the cells that maintain directional selectivity after the effect exerted by the excitotoxic agents are ganglion cells of the ON directional selective RGC type, large cells, with directional sensitivity and monostratified (sublamina ON), which respond preferentially to an slow movement of the stimulus in three directions of space [31,32,33,34], and which project preferentially on the superior colliculus or the medial terminal nucleus [35]. On the contrary, ganglion cells with directional selectivity of the ON/OFF type, (ON/OFF DS RGC) exhibit small and bistratified dendritic trees, are able to respond preferentially to rapid visual movement in four directions, and innervate the superior colliculus [36] or nuclei adjacent to the accessory optic system [37,38]. Therefore, it seems that there is a less harmful effect of NMDA and KA on the ON DS RGC compared with the ON-OFF DS RGC. 

In parallel with the decrease in cells with a directional selectivity, excitotoxic agents also affect ganglion cells without directional selectivity. After the administration of NMDA and KA, the detection of movement at certain spatial frequencies is usually the first function to be affected, as it depends on the bar stimulus that travels through the receptor fields of the neighboring ganglion cells. The loss of directional selectivity observed in the functional tests correlated with the histological disappearance observed by the immunohistochemical staining of cells that participate in directional selectivity, such as starburst amacrine cells. 

To assess the perception of visual contrast, pERG experiments were performed one week after the injection of NMDA and KA. Analysis of the pERG responses showed a decrease in amplitude (more intense in the lower frequencies (0.088, 0.120 cpd) and an increase in latency for the P50 and N95 waves, as had been suggested [39]. A parallelism was observed between the decrease in the amplitude of the pERG and the loss of ganglion cells. Although the origin of the pERG components is not exactly known, it is estimated that N95 is generated in ganglion cells. 

As the luminance in the pERG stimulus does not vary between stimuli, the pathways of activation (ON) and deactivation (OFF) by light are stimulated equally [40]. Therefore, to detect differences between these two pathways, we found it convenient to carry out another type of test: the optomotor test. To date, the optomotor response has been widely used to assess visual acuity, contrast threshold, and sensitivity to movement in laboratory animals [23,41,42]. By combining pERG measurements with the optomotor response, it is possible to assess the specific contribution of retinal cells involved in functional loss [23,43]. The decrease in response to anti-clockwise displacement optomotor stimuli is due to the preference of each eye for the detection of a direction of rotation, the left eye being the most stimulated by a light stimulus that moves clockwise, and the right eye more stimulated by stimuli moving counterclockwise. The use of degraded stimuli allowed us to observe a greater sensitivity of the OFF pathway to excitotoxic agents, as at intermediate doses of NMDA and KA, the OFF pathway of the damaged eye is completely affected, while the ON pathway is affected to a lesser extent. In view of these data, it is reasonable to think that the differences in the survival of retinal neurons are related to the expression of the different types of glutamate receptors, either NMDA type or KA type. In our opinion, it would be convenient to characterize to the greatest extent and at the cellular level the receptors of each type that retinal cells can express, in order to seek a relationship between cell survival and the maintenance of visual function, as we have tried in this work.

## 4. Material and Methods

### 4.1. Experimental Animals

A total of 50 two-month-old mice of the C57BL/6J strain were obtained from the Jackson Laboratory (Bar Harbor, ME, USA). The animals were fed ad libitum with A04 from Panlab S.L.U. (Barcelona, Spain), and were maintained in a 12:12 h circadian cycle. The mice were treated according to the European (Directiva 86/609/EEC) and Spanish regulations (Real Decreto 53/2013, 1 February 2013). The committee of research ethics and animal experimentation of the University of Alcalá approved all of the protocols. In addition, the guidelines of the association for vision and ophthalmology research were followed (The Association for Research in Vision and Ophthalmology (ARVO)). After the experiments, the animals were euthanized with a lethal overdose of a solution containing 20% sodium pentobarbital (Dolethal^®®^, Vetoquinol S.A., Lure, France) injected intraperitoneally (0.5–1 mL).

### 4.2. Intravitreal Injection of Excitotoxic Agents

The animals were anesthetized by intraperitoneal injection of a mixture of ketamine (Ketamidor, Richter Pharma AG, Wels, Austria; 100 mg/mL), xylazine (Xilagesic, CALIER, Barcelona, Spain; 20 mg/mL), and NaCl (Grifols, Barcelona, Spain; 0.9%) at a dose of 0.5 mL/150 g.

Three combinations of NMDA (6384-92-5, Sigma-Aldrich, Darmstadt, Germany) and KA (58002-62-3, Sigma-Aldrich, Darmstadt, Germany), in a concentration of 1:0.3/3:1/10:3 mM, seperately, were tested on the mice through a single intraocular dose of one microliter. The NMDA/KA solutions were injected into the right eye and one microliter of phosphate buffer saline (PBS) was injected into the left eye as a control. The intraocular injection was performed under a microdissection microscope with a cold light illumination source (Wild Heerbrugg, Intralux HE, Switzerland). One microliter-calibrated syringe (Nanofil Tm, World Precision Instruments, Sarasota, FL, USA) with a 35G needle (Nanofil Tm, NF35BV-2, World Precision Instruments) was used for the intraocular injection. In the immediate postoperative period, 2% Methocel (Ciba Vision AG, 8442 Hetlingen, Switzerland) was applied topically to the cornea to prevent corneal desiccation.

All of the immunohistochemical procedures, electrophysiological recordings, and behavioral tests on the intraocularly injected animals were performed 7 days after injection. 

### 4.3. Dose Estimation of Excitotoxic Agents 

As the doses administered to induce retinal excitotoxicity vary within the literature [16,44,45,46,47,48,49], a series of experiments were performed to determine the ideal dose for the joint administration of KA and NMDA in our model [12]. In the current experiments, the abovementioned doses were tried (NMDA/KA at 1:0.3/3:1/10:3 mM). In a series of electrophysiological experiments, only 10:3 mM NMDA/KA was chosen as a dose able to induce clear excitotoxicity.

### 4.4. Immunohistochemistry

Before enucleation, a small signal was made on the upper pole of the eye in order to ensure anatomical retinal orientation. After making an incision in the cornea, the whole eye was fixed in freshly made 4% (*w*/*v*) paraformaldehyde in 0.1 M PBS (pH 7.4) for 1 h at room temperature, and subsequently rinsed three times with PBS. Then, the anterior pole of the eye was removed, and the posterior pole was washed again in PBS. The eyeball was cryoprotected in growing concentrations of sucrose (10, 20, and 30%) diluted in 0.1M PBS (1 h for 10% and 20% and overnight for 30%). Afterward, the eyes were included in an appropriate medium for freezing (Optimal Cutting Temperature media, Sakura Finetek, CA 90501, USA) and cross sections of 14-μm thickness were made using a cryostat (Leica CM1900; Leica Microsystems, Wetzlar, Germany). The sections were mounted on Superfrost^TM^ Plus glass slides (ThermoFisher Scientific, Rockford, USA) and were stored at −20 °C until they were used for immunohistochemistry. 

Four combinations of double immunostaining were performed as follows. A rabbit monoclonal anti calbindin antibody (1:1000; CB-38a, Swant, Marly, Switzerland) and a mouse monoclonal anti-syt2b (1:50; ZNP-1, Zebrafish International Research Council, University of Oregon, Eugene, USA) were used to stain different cells in the inner retinal layers. Calbindin stains horizontal cells [50,51], some wide-field amacrine cells, and some large ganglion cells [52,53]. In rodents, the mouse monoclonal anti syt2b recognizes cone bipolar cells (OFF type) of type 2 [54,55] and type 6 [55], especially at their axon terminals (presynaptic areas). In addition, these antibodies label the presynaptic areas of horizontal cells in the mouse retina [54]. 

A goat polyclonal antibody against Brn3a (1:500; MAB1585, Merck Millipore, Darmstadt, Germany) and a rabbit polyclonal antibody against protein kinase C α (PKCα; 1:25; sc-10800, Santa Cruz Biotechnology, Santa Cruz, CA, USA) were used to stain retinal ganglion cells (RGC) and rod bipolar cells, respectively. Brn3a labels a vast majority of the mouse RGC population, even those that are injured but alive [56]. PKCα is abundantly expressed in rod bipolar cells in the mouse retina [53,57,58,59].

To immunostain different amacrine cell types, two pairs of antibodies were used. A sheep polyclonal antibody anti-tyrosine hydroxylase (TH) (1:200; AB1542, Merck Millipore) combined with a rabbit antibody anti-Dab1 (1:500; Gift from Dr. Howell, SUNY Upstate Medical University) were used to label the dopaminergic amacrine cells and their main post-synaptic neurons, the AII amacrine cells [58,60]. In addition, a goat polyclonal antibody anti choline acetyltransferase (ChAT) (1:200; AB144P, Merck Millipore), to immunostain the starburst amacrine cells, and a rabbit anti-calretinin (CR) antibody (1:500; 7679, Swant, Switzerland) to immunolabel different types of amacrine and ganglion cells, were used.

The retinal sections were incubated with the primary antibodies overnight at room temperature in a humid chamber. The next day, after three PB rinses, the samples were incubated for 1 h at RT and darkness with their matching combination of secondary antibodies at a 1:100 dilution in PB + 0.5% Triton X-100. The secondary antibodies used were donkey anti-mouse conjugated to Alexa-488 (A21202), donkey anti-mouse conjugated to Alexa-555 (A31570), donkey anti-rabbit conjugated to Alexa-488 (A21206), donkey anti-rabbit conjugated to Alexa-555 (A31572), donkey anti-goat conjugated to Alexa-488 (A11055), and donkey anti-sheep (A11015; all from ThermoFisher Scientific). Then, after three PB washes, the slides were cover-slipped with an anti-fading mounting medium (Citifluor Ltd., London, UK) and sealed with nail polish. Immunohistochemistry negative controls were conducted in parallel, omitting the primary antibody. The samples were observed in a Leica TCS SP2 confocal microscopy (Leica, Wetzlar, Germany). To image the studied cells, we used 40× and 63× oil immersion lenses, and Z projections of the maximum amplitude were taken. The same region of the temporal retina, ca. 500 um away from the optic disk, was used to take the images shown in the corresponding figures.

### 4.5. Electrophysiological Recordings

#### 4.5.1. Extracellular Multielectrode Recording

Retinal ganglion cells were extracellularly recorded from the isolated mouse retina using an array of 100 electrodes that were 1.5 mm long (inter-electrode distance = 400 µm), as described previously [61,62]. After euthanasia, the mice’s eyes were enucleated, eyeballs were hemisected, and the cornea and lens were removed under dim red illumination. Subsequently, the retinas with the pigment epithelium were carefully collected from the eyecup, mounted on a glass slide ganglion cell side up, and covered with a millipore filter. This preparation was mounted on a recording chamber; perfused with warm (36–37 °C) Ringer medium containing 124 mM NaCl, 26 mM NaCHO_3_, 22 mM glucose, 2.5 mM KCl, 2 mM MgCl_2_, 2 mM CaCl_2_, and 1.25 mM NaH_2_PO_2_; and dark-adapted for 30 min. 

A 16-bit ACER TFT monitor with a resolution of 1280 × 1024 pixels at a 60 Hz vertical refresh rate was used for the visual stimulation. Specifically, an area of 800 × 800 pixels was used for the visual stimulation. The pictures drawn on this area were projected through a beam splitter and optical lenses to be focused onto the photoreceptor layer. The retinas were flashed (full-field white) periodically, whereas the electrode array lowered slowly into the retina with the help of a Leica micromanipulator. The electrode positioning ended when a significant number of electrodes detected light-evoked single- or multi-unit responses. The retinal recordings began with a series of flashes of different intensities and moving bars of various spatial frequencies at different temporal frequencies. All of the visual stimuli were programmed in Python using the Vision Egg: an open-source library for real-time visual stimulus generation [63].

The electrode array was connected to a 100-channel amplifier (frequencies of 250 to 7500 Hz) and a digital signal processor-based data acquisition system. All of the selected channels of data, as well as the state of the visual stimulus, were digitized with a resolution of 16 bits at a sampling rate of 30 kHz with a commercial multiplexed A/D board data acquisition system (Bionic Eye Technologies, Inc., New York, USA), and were stored digitally. Neural spikes were detected by comparing the electrode signals to specifically level thresholds set for each data channel through standard procedures previously reported [64]. The supra-threshold events recorded were analyzed offline. The classification of single units was performed through a principal component analysis (PCA) method, as previously described [61,64]. Later, the assignment of every individual wave to a given cell was confirmed by analyzing the corresponding spike trains. The action potentials of each sorted unit were labeled by timestamps to generate inter-spike interval (ISI) histograms, peristimulus time histograms, and peristimulus spike plots. The following types of the stimuli were used for testing RGC functionality. 

Flash stimuli: Consists of a period of 700 ms of light at maximum intensity, followed by another of 2300 ms of darkness; the entire process was repeated 30 times. This stimulus allowed us to determine the functional cell type (ON, OFF, or ON/OFF) and the latency value. The last second of darkness prior to each light period was used to determine the basal frequency of each cell. 

Light Intensity: Consists of a period of 700 ms of light, with 11 randomly presented lighting intensities, followed by another of 2300 ms of darkness; the whole process was repeated 12 times. The sensitivity of each cell was determined by analyzing the response of the ON cells to this stimulus. 

Displacing Light Bars: A white bar at a maximum light intensity (length of 250 µm and 0.5 Hz) crossed the screen in eight different directions to determine the receptive field size and position, as well as the preferred direction of each cell. The explored directions were 0°, 45°, 90°, 135°, 180°, 225°, 270°, and 315°. The response obtained at 0° or 180° (column vector) was multiplied by the response obtained at 90° or 270° (row vector) so as to calculate the size of the receptive field. This allowed for creating a matrix, represented as an image through the Image J program, which revealed the receptive field of the cell and allowing its measurement. To determine the preferred direction, the sum vector was calculated based on the response to all directions. Once the angle of that vector was known (preferred angle), the response of the opposite angle to such stimulus was subtracted (pref - null/pref + null). Thus, an index between 0 and 1 was obtained for each cell (directional index), with 1 indicating a complete preference for a particular direction and 0 indicating any direction preferred. All of these calculations are based on the work by Elstrott and colleagues [65].

#### 4.5.2. Pattern Electroretinography

Pattern electroretinography (pERG) was used to measure the central retinal response to a constant luminance checkerboard alternating black and white [66]. A total of 18 adult mice were used in these experiments. Three experimental groups were made depending on the dose. A single dose of 1/0.3, 3/1, or 10/3 mM NMDA/KA was administered to the first (*n* = 7), second (*n* = 7), and third groups (*n* = 4), respectively. The stimulation equipment (Roland Consult, Brandenburg, Germany) consisted of a stimulator and two screens. The animals were anesthetized following the protocol previously described. After checking its state of unconsciousness through the foot reflex, its vibrissae were cut so as to avoid interference in the registry. The animals were placed on a platform raised 15 cm high and at a distance of 25 cm from both screens. The temperature of the mice was kept constant at 37° C using a closed-circuit water thermal blanket placed on the platform. A reference electrode was placed on the animal’s tongue, a ground electrode (a needle) was placed to the base of the tail, and two gold band electrodes were placed on the corneas. A few drops of methylcellulose were added (2% Methocel, Omnivision, Neuhausen, Switzerland) to protect the cornea of animals and to improve conductivity. Impedance was measured using the recording software itself so that in all cases it was below 10 KΩ. The stimulation was performed using of checkerboards configured before registration. The spatial frequencies were 0.088, 0.12, 0.17, and 0.31 cycles per degree (cpd), a frequency of change of 1 Hz, and a contrast of 100%. The stimuli were presented on two screens (919Pz, AOC, Taipei, Taiwan), with a luminance of 80 cd/m^2^ for white and 0.25 cd/m^2^ for black. A total of 400 signals were averaged to obtain an optimal result. After registration, the animals were allowed to recover in an external cage on a thermal blanket to promote awakening.

#### 4.5.3. Optomotor Response

The optomotor test is a non-invasive analysis of spatial visual acuity (or spatial frequency threshold; [23] that has proven to be efficient and reproducible [67,68,69]. The same 18 adult mice used in the pattern electroretinography were used in these experiments. A homemade Prusky style optomotor device was made [23] with four screens (FLATRON, LG, Seoul, South Korea) facing each other, forming a closed space. Inside, the awake mice were placed in the center, in a vertical transparent methacrylate cylinder. The cubicle was covered with an opaque table to prevent entering of external light, and a closed-circuit infrared camera (AVC-D5CE, SONY, Tokyo, Japan) was placed at the top of the enclosure formed by the screens, allowing the experimenter to observe the animals throughout the full experimental protocol. The four screens were connected to a computer from which the stimulus was configured.

A sequence of vertical bars with a gradient from white to black and black to white were displaced on the screens to stimulate predominantly the ON and OFF pathways, respectively. Different spatial frequencies were used (0.011, 0.022, 0.044, 0.088, 0.177, and 0.35 cpd) and they were presented in both a clockwise and anti-clockwise sense to study the effect on the left and the right eyes, respectively. In addition, these frequencies were presented at different contrasts (100, 50, 25, 10, and 5%). The stimulus was presented for 20 s, in both directions, randomly, to make the study as objective as possible. The luminance inside the optomotor device was measured to set the maximum contrast, resulting in 120 cd/m^2^ for white and 0.25 cd/m^2^ for black. 

### 4.6. Statistical Analysis

A statistical comparison of the mean in two groups was performed using the Student’s t test for normal distributions, and the Mann–Whitney U test for non-parametric distributions. The proportions between the two groups were analyzed using Chi-square. The comparison of more than one variable in two groups was carried out using two-way ANOVA and Bonferroni post hoc. They were performed using GraphPad Prism version 5.00 for Windows (www.graphpad.com, GraphPad Software, San Diego, CA, USA). Accessed May 2020.

## Figures and Tables

**Figure 1 ijms-22-06245-f001:**
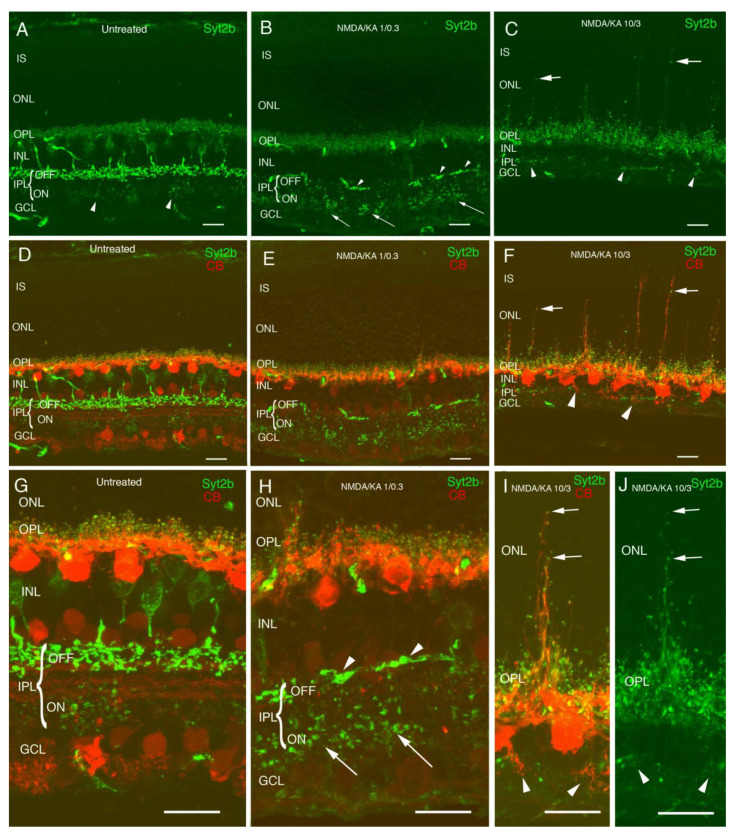
Morphological changes in OFF cone bipolar cells and horizontal cells in N-methyl-DL-aspartate (NMDA)/kainic acid (KA) treated retinas. Cryostat cross-sections immunostained with antibodies against Syt2b showing OFF and ON cone bipolar cells (**A**). Axon terminals of type 2 cone bipolar cells can be observed in the stratum of the OFF and those of type 6 cone bipolar cells stratify in the stratum of the ON showing less immunoreactivity intensity (**A**,**D** arrowheads). Double immunolabeling with calbindin antibodies shows dendrites and axon terminals of horizontal cells in the OPL (**D**,**G**). NMDA/KA 1/0.3 treated retinas show OFF cone bipolar cell death, and only a few distorted axons remain in the stratum of OFF (**B**,**E**,**H** arrowhead). In contrast, axon terminals of the ON cone bipolar cells can still be identified in the inner plexiform layer (IPL) stratum ON (**B**,**H** arrows). At this dose, the terminals of the horizontal cells show a normal morphology, and only some sprouting can be observed (**E**,**H**). NMDA/KA 10/3 treated retinas induce a clear reduction of the inner retina. While the INL, IPL, and GCL layers show a drastic thickness decrease, the IS, ONL, and OPL layer thickness are maintained, indicating that photoreceptors are not affected by the treatment (**C**,**F**). Neither ON nor OFF bipolar cells can be detected (**C**,**F**) in this condition, and horizontal cells display abnormal dendrite sprouting towards the ONL (**C**,**F**,**I**,**J** arrows) and the IPL (**C**,**F**,**I**,**J** arrowheads). IS—inner segments; ONL—outer nuclear layer; OP—outer plexiform layer; INL—inner nuclear layer; IPL—inner plexiform layer; GCL—ganglion cell layer. All of the images were obtained from temporal retina, ca. 500 µm from the optic disk. Scale bar of 10 µm.

**Figure 2 ijms-22-06245-f002:**
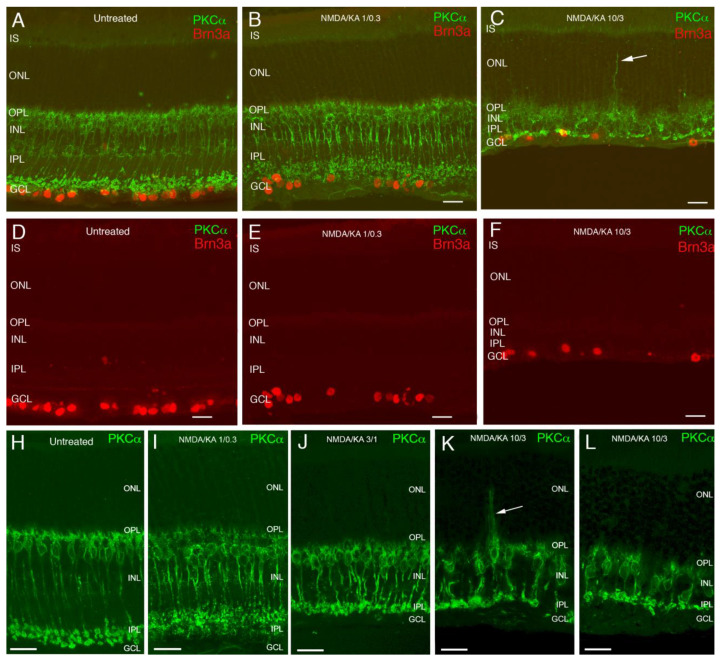
Effect of NMDA/KA treatment at different doses on ganglion cells and rod bipolar cells. Double immunolabeling with antibodies against PKC-alpha and Brn3a shows the morphology of the rod bipolar cells, with their axon terminal located in the ON stratum in the IPL (green) (**A**–**C**,**H**–**L**), and cell bodies of ganglion cells (red) (**D**–**F**). At a 1/0.3 concentration of NMDA/KA, ON rod bipolar cells are slightly shortened and a small decrease in the ganglion cell number is observed (**B**,**E**,**I**). At the 3/1 concentration of NMDA/KA, ON rod bipolar cells show an obvious shortening of their size, which accompanies the decrease in IPL thickness (**J**). At the highest NMDA/KA concentration of 10/3, a reduction in the number and size of rod bipolar cells is appreciated (**C**,**K**,**L**). Although a drastic length reduction of rod bipolar cells is obvious, these cells retain their dendrites and axons, adapting their morphology to the reduction in size of the IPL (**K**,**L**). Some of the bipolar cells present sprouting of their dendrites towards the ONL (**C**,**K**, arrow). IS—inner segments; ONL—outer nuclear layer; OPL—outer plexiform layer; INL—inner nuclear layer; IPL—inner plexiform layer; GCL—ganglion cell layer. Scale bar of 20 µm.

**Figure 3 ijms-22-06245-f003:**
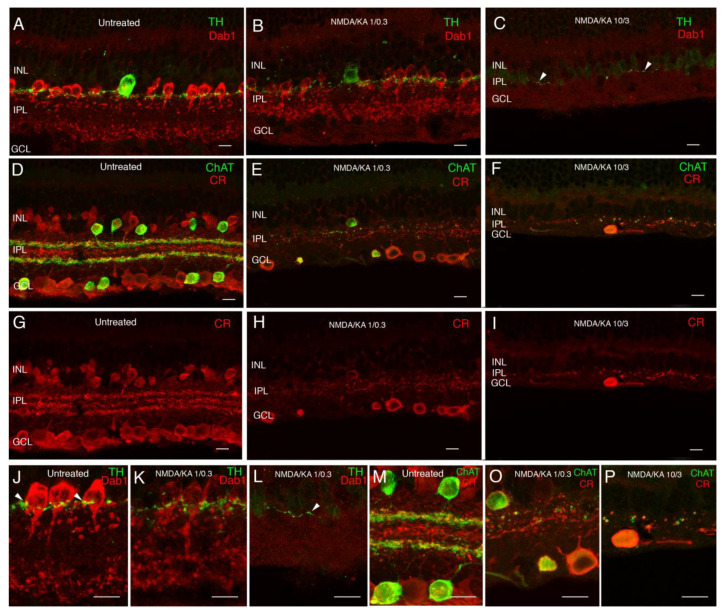
Response of amacrine cells to excitotoxicity. Double immunolabeling with antibodies against tyrosine hydroxylase (TH) and Dab1 (**A**–**C**,**J**–**L**). Tyrosine hydroxylase shows the dopaminergic amacrine cells and their dendritic plexus in the S1 stratum in the IPL (**A**,**J**, green). Dab1 shows the AII amacrine cells, whose typical lobular appendages are mainly in the OFF layer and their dendritic terminals in the ON layer (**A**,**J** red). Synaptic contacts from dopaminergic cells around the cell bodies of AII amacrine cells can be observed (**J** arrowheads). A decrease in TH and Dab1 immunoreactivity intensity is found in response to the 1/0.3 concentration of NMDA/KA. The morphology of AII amacrine cells looks disorganized, but the synaptic contacts with the dopaminergic cells still remain (**B**,**K**). At the NMDA/KA 10/3 dose, AII amacrine cells cannot be identified and only a few dendrites of dopaminergic cells can be observed in the S1 stratum of the IPL (**C**,**L** arrowhead). Double immunolabeling with antibodies against calretinin and choline acetyltransferase (**D**–**I**,**M**–**P**). Calretinin immunoreactivity labels several types of amacrine cells and ganglion cells with three typical plexuses of dendrite stratification in the IPL (**D**,**G**,**M** red). ChAT immunoreactivity is found in starburst amacrine cells, whose cell bodies are located in the INL and in the ganglion cell layer, and their dendrites stratify in two specular plexuses in the ON and OFF layers of the IPL (**D**,**M** green). At the 1/0.3 concentration of NMDA/KA, calretinin immunoreactive amacrine cells cannot be identified, and only a few ChAT amacrine cells and some CR ganglion cells remain (**E**,**H**,**O**). Both plexuses experience a big disruption and disorganization. At a 10/3 concentration, only some spots of ChAT and CR immunoreactivity can be observed in the IPL, accompanying IPL degeneration (**F**,**I**,**P**). INL—inner nuclear layer; IPL—inner plexiform layer; GCL—ganglion cell layer. Scale bar of 10 µm.

**Figure 4 ijms-22-06245-f004:**
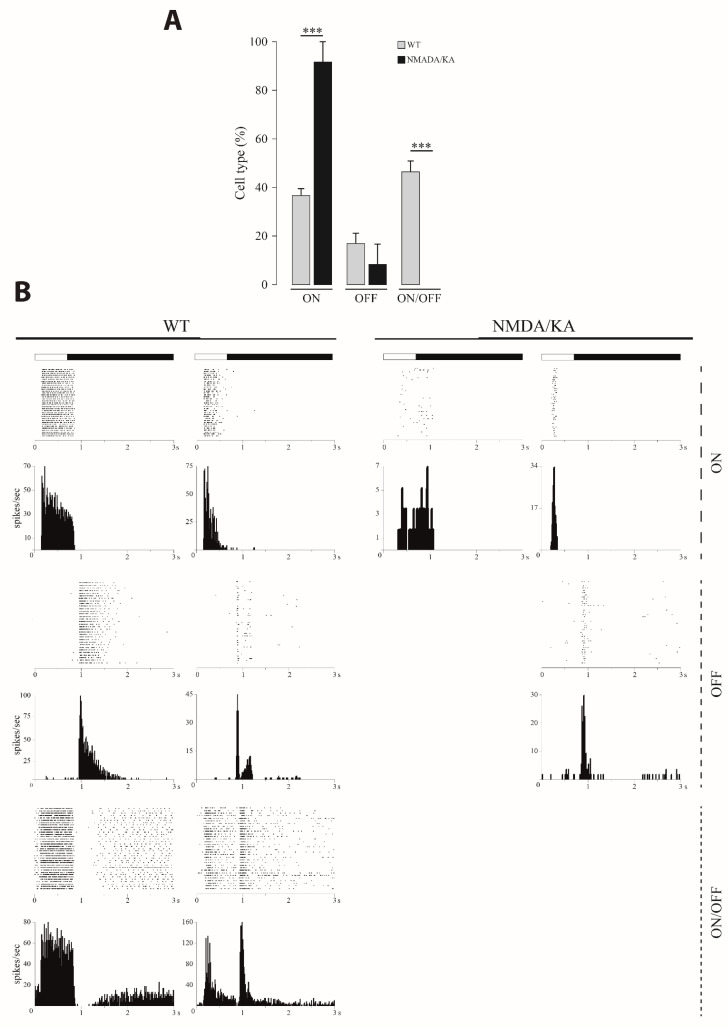
Quantification of the different retinal ganglion cells (RGC) types responses. (**A**). Percentages of the different functional types of RGC (ON, OFF, and ON/OFF) recorded in the control/wild type (WT retinas (*n* = 4) and those injected with 10:3 mM NMDA/KA (*n* = 4; t-test; *p* ≤ 0.001 for ON and ON/OFF cells). A significant decrease in the number recorded cells was observed for the NMDN/KA injected retinas (t-test; *p* = 0.018). The total number of ON, OFF, and ON/OFF cells were 89, 35, and 113 for the WT retinas and 48, 2, and 0 for NMDA/KA injected eyes, respectively. Data are presented as mean ± SEM. (**B**). Examples of sustained (left) and transient (right) post stimulus raster plot of light responses are represented for each RGC type (ON, OFF, and ON/OFF) for both retinas from the WT/control animals and those injected with NMDA/KA. The light stimulus is represented above the plots. A corresponding cumulated post stimulus time histogram (PSTH) for all recording examples is also shown for all RGC types. The firing frequency evoked by the light stimulus is greatly reduced in the NMDA/KA group. (***, *p* ≤ 0.001).

**Figure 5 ijms-22-06245-f005:**
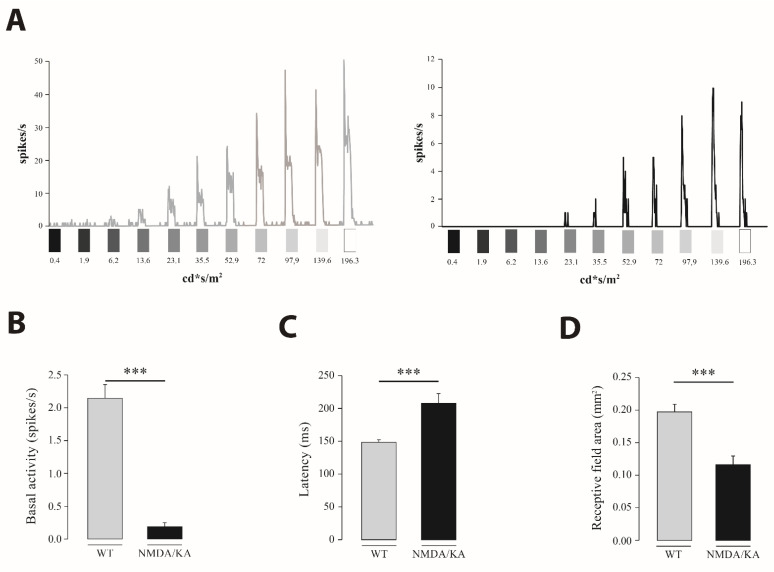
Functional characterization of RGC in the control and injured group. (**A**). Examples of light responses recorded from RGC to increasing light intensities of an ON-type cell from the control (left) and NMDA/KA (right) retinas. The left post stimulus time histogram (PSTH) corresponds to an RGC from the control retina, which increases the firing rate as the light intensity increases, being sensitive to 6.2 cd·s/m^2^ intensity. However, on the right, RGC from the NMDA/KA injected eyes needs brighter light intensities to evoke light responses. A wide reduction in firing frequency in the RGC from injured retina is also noted. (**B**). Differences in the spontaneous basal activity recorded from the RGC in control and NMDA/KA injected eyes. A statistically significant difference is observed between both animal groups (Mann–Whitney U; *p* ≤ 0.001). (**C**). Histogram representation of the mean latency of the ON and ON/OFF cell types (only the ON phase) for the control group (*n* = 202) and just the ON type in the NMDA/KA injected group (*n* = 48), showing statistically significant differences (Mann–Whitney U; *p* ≤ 0.001). (**D**). Comparison of the receptive field size of RGC in the control and NMDA/KA injected retinas. A statistically significant decrease was observed in the NMDA/KA when compared with the control retinas (Mann–Whitney U; *p* ≤ 0.001). The receptive field size is measured by drawing the matrix obtained by multiplying the response after light bars stimulation in two orthogonal directions for each cell in Image J. Data are presented as mean ± SEM. (***, *p* ≤ 0.001).

**Figure 6 ijms-22-06245-f006:**
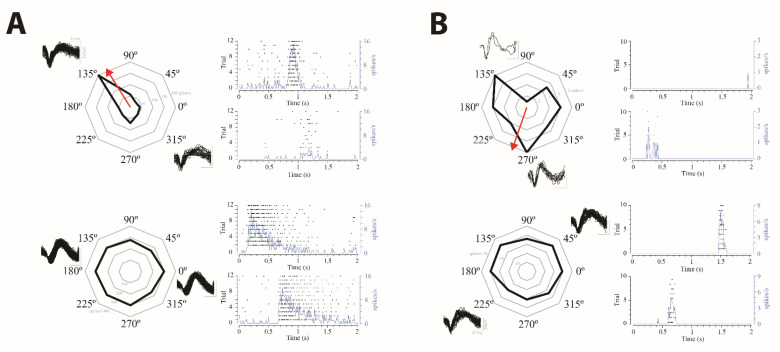
Directional selectivity in ON RGC. Examples of ON-RGC with directional selectivity (up) or without directional selectivity (down) recorded from the control retinas (**A**) and retinas from the NMDA/KA injected eyes (**B**). Each cell example includes a radial plot of the spike rate response to motion in eight directions across ten to twelve repetitions and the spike waveform, and the post stimulus time raster plot and cumulative recording line for the response to preferred (above) and the null (below) directions is also shown, (**B**). The red arrow indicates the preferred direction calculated as the vector sum of the response. Directional selective RGC from the NMDA/KA injected retinas shows an apparent broader response in the preferred direction than RGC from the control retinas.

**Figure 7 ijms-22-06245-f007:**
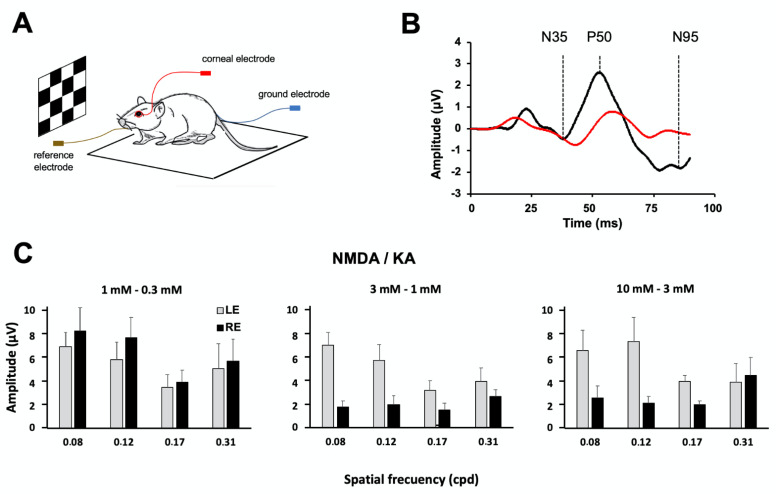
Effect of NMDA/KA injection on retinal ganglion cell functionality evaluated by pattern electroretinography (pERG). (**A**). Schematic diagram of the pERG test experimental design (**B**). Example of the pERG trace recorded before the injection of 10:3 mM NMDA/KA (black line) and one week after the injection (red line). Before injection, the trace recording shows the three characteristic components of the pERG (N35, P50, and N95). However, one week after the injection, just the P50 component is clearly distinguished. (**C**). Histogram representations of the N95 pERG component amplitude, averaged (mean ± SD) from the right eye (RE) and the left eye (LE) at different spatial frequencies. The right eyes of the animals are injected with NMDA/KA at 1:0.3 mM (*n* = 7), 3:1 mM (*n* = 7) and 10:3 mM (*n* = 4). The control left eyes are injected with PBS. A statistically significant decrease in the N95 amplitude is observed after eye injection with 3:1 mM and 10:3 mM NMDA/KA (*p* < 0.05, two way ANOVA).

**Figure 8 ijms-22-06245-f008:**
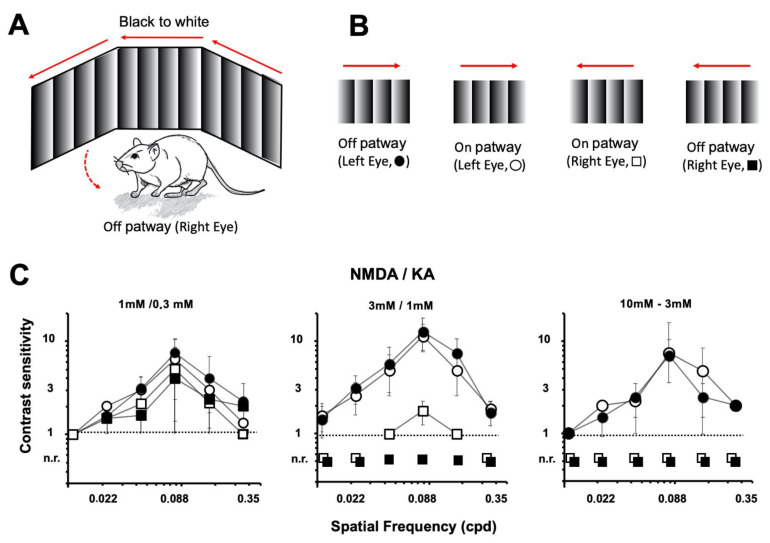
Effect of NMDA/KA injection on visual acuity. Optomotor test evaluation of visual acuity (contrast sensitivity) in experimental animals injected intraocularly into the right eye with NMDA/KA (1:0.3 mM, *n* = 7; 3:1 mM, *n* = 7; and 10:3 mM, *n* = 4). The control left eye is injected with PBS. (**A**). Schematic diagram of the optomotor test experimental design. (**B**). The OFF visual pathway is tested by vertical bars with a horizontal gradient from white to black (closed symbols). The ON visual pathway is tested by vertical bars with a horizontal gradient from black to white (open symbols). Clockwise (circles) and counterclockwise directions (squares) are tested. The spatial frequencies of the moving visual stimuli are tested at different spatial frequencies. (**C**). Animals fail to follow the visual stimulus in an anticlockwise direction (mediated by the injured right eye), when compared with the clockwise response (mediated by the control left eye) as a result of NMDA/KA excitotoxicity at 3:1 mM or 10:3 mM (*p* < 0.05, two tail ANOVA). For 3:1 mM NMDA/KA injection, animals are still able to detect light stimuli mediated by the ON pathway, while the OFF pathway is completely insensitive (n.r.—no response to stimuli).

## Data Availability

Not applicable.

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
