# Peer review of "Visual Disfunction due to the Selective Effect of Glutamate Agonists on Retinal Cells"

_ijms, 2021, doi:10.3390/ijms22126245_

Round 1

Reviewer 1 Report

The manuscript “VISUAL DISFUNCTION DUE TO SELECTIVE EFFECT OF 2 GLUTAMATE AGONISTS ON RETINAL CELLS.” By Milla-Navarro et al. investigates the cell type selectivity of glutamate toxicity in the retina. The authors use a mixture of NMDA and kainic acid to induce neurotoxicity. The stated importance of this study is to understand the importance of glutamate receptors to different types of neurons. Cells and pathways most resistant to excitotoxicity using 2 month old rats. Ideal dose was reported previously in Calvo et al. This manuscript is long, and the discussion could be shortened.

Specific comments:

The timing of injection to each test, and harvest/enucleation time post injection needs to be clearly presented.

The use of multiple doses with for comparison to the vehicle treated left eye is a strength.

Figures 1, 2 and 3 provide convincing histology and cell-type specific staining.

Quantification and relative abundance of various on, off and on/off cells is a strength.  

Biological activity measured in response to light at different intensities is very good.

Data and presentation are strong, but discussion could be shortened.

In general, the manuscript is well written, the results presented and interpreted fairly.

I found one proof reading error in the introduction:

“In the case of KA, it has been 53 shown that a low concentration of this molecule is capable of causing intracellular calcium 54 elevation and the death of neural cells without significant depolarization ({ HYPERLINK 55 "https://www.ncbi.nlm.nih.gov/pubmed/?term=Shen%20W%5BAuthor%5D&cau-56 thor=true&cauthor_uid=11877532" }{ HYPERLINK "https://www.ncbi.nlm.nih.gov/pub-57 med/?term=Slaughter%20MM%5BAuthor%5D&cauthor=true&cauthor_uid=11877532" }.

Author Response

The sentence related to “KA effect on Calcium concentration” indicated by the reviewer #1 has been checked and modified accordingly (lines 56 – 58).

According to the reviewer #1 request, the time between intraocular injection of excitatory agents and the day when experiments were performed is now properly included into the method section (lines 130 – 132)

The discussion has been shortened as suggested by reviewer #1. More than 60 lines of text, and their corresponding cites have been removed. (about one third of the discussion text).

English revisions, our manuscript has been checked by a native English-speaking researcher.

Reviewer 2 Report

In the present study, a mixture of the glutamate analogues NMDA and kainic acid (KA) are injected into the eyes of mice at varying concentrations to study neurotoxic effects on the retina. The other eye was injected with saline and used as a control. A variety of methods are used including immunofluorescence, ERG, electrophysiological recordings and visual function measurement. The main finding is that the highest dose of both drugs has the strongest effect on structure and function and that the OFF pathway is more strongly effected than the ON pathway.

MAJOR POINTS:

  1. The abstract should highlight the new findings of the present study.
  2. The manuscript is rather lengthy, contains a large number of grammatical oddities, errors (some are listed below) and inconsistencies making it hard to read. It requires careful revision. Please also carefully check for repetition (e.g. the first paragraph of the result section is a repetition of what was already said in the methods section).
  3. The neurotoxic effects of NMDA and KA on the retina have been studied extensively previously but in most cases either the effect of NMDA or the effect of KA were addressed. The rationale of applying NMDA and KA together should be explained better. Moreover, the discussion should compare single application with mixed application more clearly.
  4. The study is a continuation of a previous study by the same group and some of the results confirm previous results (e.g. see Figure 6 in Calvo et al. 2020 and Figs. 1 and 2 in the present study). It should be clearly stated throughout which findings are confirmation of previous findings and which findings are new. Also please state clearly in case the same animals as in the previous study were used.
  5. It is not clear which retinal regions were used in the various experiments.
  6. The left eye of each animal was injected with saline and thus served as control, yet this eye is often referred to as “untreated” and the right eye is referred to as “injected” eye. In other cases the term “injected animals” is used. As far as I can tell normal or untreated eyes were not investigated in the present study.
  7. The discussion is way too long and hard to follow. It contains a lot of repetition of the results and it is not always clear what point is being made. Specifically, the question of how the present results compare to separate application of NMDA and KA needs to addressed.

Other points:

None of the hyperlinks seems to be working.

Please check all numbers for example “ 0,9” should be “0.9” in English

For calbindin labelling of the mouse retina Haverkamp and Wässle (2000) is a better reference than Mills and Massey who investigated rabbit retina. 

For disabled cite Rice SR, Curran T. 2000. Disabled-1 is expressed in type AII amacrine cells in the mouse retina. J Comp Neurol 424:327–338

For PKC labelling of rod bipolar cells in rodents cite Greferath et al., 1990 and Haverkamp and Wässle, 2000

Calretinin immunoreactivity is found in a variety of amacrine cells in mouse retina (Haverkamp and Wässle, 2000), but not in the AII cell as claimed (line 316)

Figure 1 is nice but the data are poorly described in the result section. Please refer to the appropriate panels within the text. Also please state whether all panels were taken from comparable retinal regions e.g. nasal or temporal retina at such and such distance from the optic disk.

Figure 4: Please give the numbers of cells recorded.

Figure 7: Please check legend

Author Response

A new sentence hi¡ghlighting the new findings has been included into the abstract, as requested by the reviewer #2 (lines 36 – 38).

The rationale of applying NMDA and KA together is better explained in the introduction, as suggested by the reviewer #2 (lines 61 – 63). It was also explained in the discussion (lines 594 – 598).

Previous studies showed the deleterious effect of NMDA and KA, when used at high concentrations (30 mM NMDA and 10 mM KA). In the present study we have used lower concentrations of both agents, in order to demonstrate from which concentration a harmful effect can be observed, and which cells or pathways are more sensitive to excitotoxic agents.

Our findings confirm previous observations, and further demonstrate that lower concentrations of excitatory agents are able to induce stronger effect on the OFF pathway than in the ON pathway. This fact It is now stated in the text (line 96), and better explained in the shortened discussion.

No any result obtained form our previous work (Calvo et al., 2020) is used for the present work. All results shown on present study have been obtained from new series of specifically designed experiments, since lower drug concentrations were used and now it is stated on line 96 of the present manuscript.   

The region of the retina where the photographs were taken is now indicated in the text. The same region of the temporal portion of the retina, about 500 um from the optic disk, was used to take all the images shown in figures 1, 2 and 3 (lines 193 - 194).

We agree that the term “injected animal” is confusing. So, all changes have been made throughout the text to indicate the “injected eye” (lines 411, 412, 432, 448, 450, 473).

The discussion text now compare the single application of the NMDA and KA with mixed application of both glutamate agonists (lines 585 – 591), as requested by reviewer #2

We did not use hyperlinks into the text. Editorial must check this.

All numbers have been checked along the text and corrected accordingly.

Staining of AII amacrine cells by calretinin has been removed from the text as indicated by reviewer #2.

The discussion has been shortened as suggested by reviewer #2. One third of the original manuscript discussion test and the corresponding cites have been removed. 

The paragraph describing Figure 1 (lines 313 – 338) has been modified accordingly to suggestion made by reviewer #2, and now the appropriate panels are indicated within the text. The area of the retina from where the images were taken is now indicated in the figure legend

The number of cells recorded of each physiological class (ON, OFF and ON/OFF) from each experimental group (control vs. NMDA/KA) is now indicated in the figure legend, as requested by the reviewer #2 (lines 424 – 426).

Legends of figure 7 (lines 494 – 502) and figure 8 (lines 531 – 540) have been checked and rewritten as suggested by reviewer #2. Now, the figure legends describe quite properly the schemes and graphics shown in the figures.

New references have been included as suggested by the reviewer #2:

  • Greeferath et al., 1990
  • Haverkamp and Wässle, 2000
  • Rice and Curran, 2000

Since the whole English has been checked and rewritten and the discussion has been shortened, as suggested by both reviewers, new references have been included and some references have been removed from the reference list.

Finaly, since both referees suggested the manuscript to undergo extensive English revisions, our manuscript has been checked by a native English-speaking researcher.